# An Integrated Model for Transformer Fault Diagnosis to Improve Sample Classification near Decision Boundary of Support Vector Machine

**Yiyi Zhang** [1] , **Yuxuan Wang** [1], **Xianhao Fan** [1], **Wei Zhang** [2], **Ran Zhuo** [3], **Jian Hao** [4] and **Zhen Shi** [1,*]

1   Guangxi Key Laboratory of Power System Optimization and Energy Technology, Guangxi University, Nanning 530004, China; yiyizhang@gxu.edu.cn (Y.Z.); 1912301053@st.gxu.edu.cn (Y.W.); 1812301008@st.gxu.edu.cn (X.F.)
2   Guangxi Power Grid Co., Ltd. Electric Power Research Institute, Nanning 530000, China; zhang_w.sy@gx.csg.cn
3   Electric Power Research Institute of China Southern Power Grid Company Limited, Guangzhou 510080, China; zhuoran@csg.cn
4   School of Electrical Engineering, Chongqing University, Chongqing 400030, China; haojian2016@cqu.edu.cn
*   Correspondence: z.shi@gxu.edu.cn

**Abstract:** Support vector machine (SVM), which serves as one kind of artificial intelligence technique, has been widely employed in transformer fault diagnosis when involving dissolved gas analysis (DGA). However, when using SVM, it is easy to misclassify samples which are located near the decision boundary, resulting in a decrease in the accuracy of fault diagnosis. Given this issue, this paper proposed a genetic algorithm (GA) optimized probabilistic SVM (GAPSVM) integrated with the fuzzy three-ratio (FTR) method, in which the GAPSVM can judge whether a sample is near the decision boundary according to its output probabilities and diagnose the samples which are not near the decision boundary. Then, FTR is used to diagnose the samples which are near the decision boundary. Combining GAPSVM and FTR, the integrated model can accurately diagnose samples near the decision boundary of SVM. In addition, to avoid redundant and erroneous features, this paper also used GA to select the optimal DGA features. The diagnostic accuracy of the proposed GAPSVM integrated with the FTR fault diagnosis method reached 86.80% after 10 repeated calculations using 118 groups of IEC technical committee (TC) 10 samples. Moreover, the robustness is also proven through 30 groups of DGA samples from the State Grid Co. of China and 15 practical cases with missing values.

**Keywords:** power transformer; fault diagnosis; probabilistic support vector machine; expert experience; dissolved gas analysis feature; genetic algorithm

## 1. Introduction

Oil-immersed power transformers are important pieces of power transmission equipment in the power system. Transformer failure causes widespread power blackout, resulting in economic losses that cannot be estimated [1–3]. Therefore, the safe and stable operation of the transformer is important to the power system, and it is of great importance to diagnose transformer faults such as over-heating and discharges in time and correctly.

In the existing research, dissolved gas analysis (DGA) has been widely used as the on-line fault monitoring approach for power transformer fault diagnosis. The gases dissolved in the transformer oil mainly include hydrogen ($H_2$), methane ($CH_4$), acetylene ($C_2H_2$), ethylene ($C_2H_4$), and ethane ($C_2H_6$)

from oil decomposition, in conjunction with carbon monoxide (CO) and carbon dioxide ($CO_2$) from paper decomposition. Currently, the commonly used DGA fault diagnosis methods, such as the Roger ratio [4], International Electrotechnical Commission (IEC) three ratios [5], and Doernerburg ratio [6], are based on experimental experiences, which results in many problems in application. For example, the Rogers ratio reflects the thermal decomposition temperature range only, and IEC three ratios has incomplete coding [7–9].

With the rise of artificial intelligence (AI) technology, the fault types of transformers can be diagnosed by complex function mapping based on the DGA data [10–13]. However, DGA benchmarking samples are difficult to acquire. Large-scale models like deep learning are challenging to apply to DGA-based fault diagnosis. Support vector machine (SVM) performs well with small samples and has a strong generalization ability [14,15]. Hence, SVM is widely used in transformer fault diagnosis based on DGA. Previous research [16] proposed an SVM-based transformer fault diagnosis method. Using the characteristic gas of DGA as the input feature of SVM, the fault type of the transformer can be diagnosed through the trained model. In order to reduce redundancy and incorrect input features, the authors of [17] proposed a genetic programming method to select effective DGA features to improve diagnostic accuracy. Moreover, the authors of [18] proposed a genetic algorithm (GA) combined with SVM to select the optimal DGA gas ratios to improve the diagnostic accuracy. Additionally, the kernel parameter and slack variable of SVM should be set manually, as inappropriate parameter settings reduce the accuracy of fault diagnosis. Thus, combining feature selection and SVM parameter optimization, the authors of [19] proposed improved krill herd (IKH) optimized SVM (IKHSVM), in which IKH can optimize the internal parameters of SVM. To avoid the noise and outliers affecting the diagnostic accuracy, the authors of [20] proposed fuzzy SVM, which can reflect the impact of different samples on SVM by assigning weights to them. The weights assigned to noise and outliers are reduced, which has little impact on the model. Moreover, to avoid the limitations of single SVM, the authors of [21,22] introduced SVM and another three classifiers combined into an ensemble classifier, and this method could always select the most accurate classifier using multi-objective Particle Swarm Optimization algorithm. In addition, the authors of [23] also proposed an association rule mining method, which can select the most appropriate fault diagnosis method from two empirical rules and three AI-based classifiers. These integrated methods have significantly improved the diagnostic accuracy. However, these methods only combine several classifiers and select the most effective classifier for diagnosing transformer fault types according to certain rules or optimization algorithms, resulting in a large time complexity in the calculation process. In addition, these studies have not pointed out the defects of each classifier.

According to previous research [24], SVM is prone to misclassifying certain samples located near the decision boundaries. Therefore, the key to improving the classification performance of SVM is to effectively classify the samples near the decision boundary. The authors of [25] proposed the probabilistic SVM (PSVM), which provides the probability of each class. It can be judged whether the sample is near the decision boundary according to the output probability of PSVM. To effectively diagnose the samples near decision boundaries and reduce the complexity of the integrated model, this paper introduced the expert experiment-based fuzzy three-ratio (FTR) model [26], which is not influenced by whether a sample is near the decision boundary of the SVM. Combining PSVM and FTR, a transformer fault diagnosis approach based on GA optimized probabilistic SVM (GAPSVM) integrated with FTR is achieved. The integrated model improves the diagnostic accuracy by effectively diagnosing the samples near the decision boundaries of SVM. Thus, the proposed approach has not only the superiority of AI-based algorithms but also combines with the expert experience to eliminate the impact of data quality on AI-based algorithms. Moreover, taking into account the redundant or wrong features, this paper also uses GA to screen the optimal DGA features (ODF) from 36 groups of generated features.

## 2. A Fault Diagnosis Approach Based on GAPSVM Integrated with Expert Experience

### 2.1. Optimization of Transformer DGA Features Based on GA and SVM

#### 2.1.1. Gas Features Dissolved in Oil

The conventional DGA features mainly include $H_2$, $CH_4$, $C_2H_2$, $C_2H_4$, $C_2H_6$, CO, $CO_2$, and total hydrocarbon (TH). To find the ODF, contents of the above gases and the ratio of every two gas contents formed all DGA features to be selected. The corresponding DGA features are numbered in Table 1. No.1–No.8 are the conventional DGA features, and No.9–No.36 are the ratios of every two gas contents. ODF is selected from the above DGA features.

**Table 1.** DGA features to be selected.

| No | DGA Feature | No | DGA Feature | No | DGA Feature |
|----|-------------|----|-------------|----|-------------|
| 1 | $H_2$ | 13 | $H_2/CO$ | 25 | $C_2H_2/CO_2$ |
| 2 | $CH_4$ | 14 | $H_2/CO_2$ | 26 | $C_2H_2/TH$ |
| 3 | $C_2H_2$ | 15 | $H_2/TH$ | 27 | $C_2H_4/C_2H_6$ |
| 4 | $C_2H_4$ | 16 | $CH_4/C_2H_2$ | 28 | $C_2H_4/CO$ |
| 5 | $C_2H_6$ | 17 | $CH_4/C_2H_4$ | 29 | $C_2H_4/CO_2$ |
| 6 | CO | 18 | $CH_4/C_2H_6$ | 30 | $C_2H_4/TH$ |
| 7 | $CO_2$ | 19 | $CH_4/CO$ | 31 | $C_2H_6/CO$ |
| 8 | TH | 20 | $CH_4/CO_2$ | 32 | $C_2H_6/CO_2$ |
| 9 | $H_2/CH_4$ | 21 | $CH_4/TH$ | 33 | $C_2H_6/TH$ |
| 10 | $H_2/C_2H_2$ | 22 | $C_2H_2/C_2H_4$ | 34 | $CO/CO_2$ |
| 11 | $H_2/C_2H_4$ | 23 | $C_2H_2/C_2H_6$ | 35 | $CO/TH$ |
| 12 | $H_2/C_2H_6$ | 24 | $C_2H_2/CO$ | 36 | $CO_2/TH$ |

#### 2.1.2. DGA Feature Selection Based on GA Combined with SVM

Feature engineering is an important procedure in machine learning. Redundant features will reduce the calculation speed of the algorithm, and incorrect features may reduce the accuracy of the algorithm [27]. The feature selection method based on GA and SVM proposed in [28] is improved and used in this work for ODF selection; the binary encoding of chromosomes is shown in Figure 1.

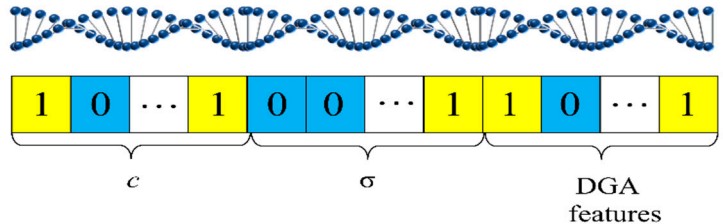

**Figure 1.** The binary encoding of chromosomes.

The chromosomes of GA are generated by binary coding. Each chromosome consists of three genes. The first two genes are penalty factor $c$ and $\sigma$ of SVM; the third gene is the 36 sets of DGA features in order. Moreover, the corresponding relationship is shown in Figure 1. The encoding "1" represents the DGA feature that has been selected, while "0" represents the one which has not been selected. The parameter settings of GA are shown in Table 2. The ODF can be obtained by GA iterations using k-fold cross-validation (CV) accuracy as the fitness function.

**Table 2.** Parameter settings of GA.

| Parameters | Settings |
|---|---|
| Maximum iteration | 200 |
| Population size | 100 |
| Crossover probability | 0.9 |
| Mutation probability | 0.01 |
| Range of $C$ | [0, 200] |
| Range of $\sigma$ | [0, 100] |

### 2.1.3. Nonlinear Support Vector Machine

The conventional SVM is a linear and two-class classifier which must be upgraded as the transformer fault diagnosis is a nonlinear and multi-classification problem. The nonlinear SVM model and its flowchart are shown in Figure 1.

$$\min\Phi(\omega,\xi) = \tfrac{1}{2}\|\omega\|^2 + C\sum_{i=1}^{l}\xi_i$$
$$s.t.\begin{cases} y_i\left[\omega^T\varphi(x_i) + \lambda\right] \geq 1 - \xi_i \\ \xi_i \geq 0,\ i = 1,2,\ldots,l \end{cases} \tag{1}$$

where $\xi_i$ is a slack variable and $C$ is a penalty factor. The Lagrange function is presented as follows:

$$L(\omega,\lambda,\xi,\alpha,\beta) = \Phi(\omega,\xi) -$$
$$\sum_{i=1}^{l}\alpha_i\left\{y_i[\omega^T\varphi(x_i) + \lambda] - 1 + \xi_i\right\} - \sum_{i=1}^{l}\beta_i\xi_i \tag{2}$$

Additionally, the decision function is:

$$y = sign[\sum_{i=1}^{l}\alpha_i y_i K(x, x_i) + \lambda] \tag{3}$$

where $K(x_i, x_j)$ is the kernel function which maps low-dimensional space to high-dimensional space. The commonly used kernel functions are Gaussian radial basis functions (RBF), polynomial functions, etc. There is only one parameter to be fitted in RBF function. Therefore, RBF is used as the kernel function of SVM:

$$K(x_i, x_j) = \exp(-\frac{\left\|x_i - x_j\right\|^2}{2\sigma^2}) \tag{4}$$

### 2.1.4. Probabilistic SVM

To output the probability of each class, Platt [25] proposed a sigmoid-fitting method to obtain probabilistic outputs for SVM instead of uncalibrated values.

$$p_i = \frac{1}{1 + exp(Af_i + B)} \tag{5}$$

where $f_i$ is the sample's unthresholded output, $y_i$ is the sample's label, $A$ and $B$ are the parameters to be fitted by minimizing a cross-entropy function of $p_i$ and $t_i$, which is shown in Equation (6). $t_i$ is the target probabilities, which is defined as Equation (7).

$$\min -\sum_{i} t_i log(p_i) + (1 - t_i)log(1 - p_i) \tag{6}$$

$$t_i = \frac{y_i + 1}{2} \tag{7}$$

*2.2. GAPSVM Integrated with FTR Model*

2.2.1. Fuzzy Three-Ratio Model

For conventional IEC three ratios for transformer fault diagnosis, ratios of $C_2H_2/C_2H_4$, $CH_4/H_2$, and $C_2H_4/C_2H_6$ are respectively encoded in a certain interval; the coding rule of the three-ratio method is shown in Table 3. The fault types can be recognized according to the corresponding codes in Table 4.

**Table 3.** Coding rule of three-ratio method.

| Ranges of Gas Ratios | Codes of Different Gas Ratios | | |
| --- | --- | --- | --- |
| | $C_2H_2/C_2H_4$ | $CH_4/H_2$ | $C_2H_4/C_2H_6$ |
| <0.1 | 0 | 1 | 0 |
| 0.1–1 | 1 | 0 | 0 |
| 1–3 | 1 | 2 | 1 |
| >3 | 2 | 2 | 2 |

**Table 4.** Fault types of DGA codes.

| No | Fault Type | Code of the Ratios | | |
| --- | --- | --- | --- | --- |
| | | $C_2H_2/C_2H_4$ | $CH_4/H_2$ | $C_2H_4/C_2H_6$ |
| 1 | Discharge of low energy density | 1 or 2 | 0 | 1 or 2 |
| 2 | Discharge of high energy density | 1 | 0 | 2 |
| 3 | Thermal fault of low temperature < 300 °C | 0 | 0 or 2 | 1 or 2 |
| 4 | Thermal fault of high temperature ≥ 300 °C | 0 | 2 | 1 or 2 |
| 5 | No fault | 0 | 0 | 0 |

However, the coding boundaries are too clear and depend heavily on the experience; a very small increase in the gas ratio may sharply change the codes. In fact, the boundaries of each code should be fuzzy [29].

In the FTR model, IEC codes 0, 1, 2 are replaced by ZERO, ONE, TWO; each gas ratio can be represented by a fuzzy vector. $[u_{ZERO}(r_i), u_{ONE}(r_i), u_{THREE}(r_i)]$ is used to replace the IEC codes to obtain the fuzzy boundaries, where $r_1 = C_2H_2/C_2H_4$, $r_2 = CH_4/H_2$ and $r_3 = C_2H_4/C_2H_6$. $u_{ZERO}(r_i)$, $u_{ZERO}(r_i)$, $u_{ZERO}(r_i)$ are membership functions. In previous studies on the fuzzy three-ratio model, the triangular membership function is often used, because the triangular membership function has fewer parameter settings and the sine curve transition is relatively smooth. Therefore, the triangular membership function is also chosen in this paper. Replace the conventional logic "AND" by "min", "OR" by "max", then calculate the fuzzy fault diagnosis vector $f(i)$ [30]. To make the sum of $f(i)$ equal to one, the normalization is shown as Equation (8).

$$f'(i) = \frac{f(i)}{\sum\limits_{j=1}^{5} f(j)} \tag{8}$$

According to Equation (8), if $f'(i)$ is the maximum, it can be considered that the transformer has No.*i* fault. If the second maximum $f'(j)$ is very close to $f'(i)$, the transformer is considered to have both No.*i* and No.*j* fault.

2.2.2. Analysis of PSVM and the Combination Method of GAPSVM and FTR

The outputs of the GAPSVM are the probabilities of each fault type of a sample; $p_i$ represents the probability of the No. *i* fault. Thus, there might be the following conditions:

- If $p_i > 0.5$, the SVM has high confidence that the sample belongs to the corresponding fault type.
- If $p_i \leq 0.5$, the sample is near the decision boundary of the SVM which carries out the classification of the fault in this situation. SVM has low confidence to classify the samples, and misclassification usually occurs in this situation.
- The sample is more likely to be divided into the class with higher probability.

Based on the above theory, if the maximum probability of each classification does not reach 0.5, it is considered that SVM is not sufficient for the sample and the FTR model will be chosen for fault diagnosis.

The flowchart of the GAPSVM integrated with FTR model is shown in Figure 2. The ODF is selected by GA combined with SVM, and DGA samples are divided into a training set and testing set. Afterwards, the GAPSVM gives the probabilities of each fault type. If the maximum probability exceeds 0.5, the diagnosis result will be given by GAPSVM; otherwise, the sample will be diagnosed by FTR. The total equation for the integrated model is given as Equation (9). $P_i$ is the max GAPSVM output probability of a certain sample. When $P_i$ is less than or equal to 0.5, the sample is considered to be near the decision boundary of SVM, so it is diagnosed by FTR; $u(r_i)$ is the fuzzy vector of FTR. FTR calculates the fuzzy vectors according to max and min and finally diagnoses the fault type of the sample. When $P_i$ is more than 0.5, the sample is considered not near the decision boundary of SVM, and the sample is diagnosed by GAPSVM. Where $\xi_i$ is a slack variable and $C$ is a penalty factor, and $K(x_i, x_j)$ is the kernel function, $L(\omega, \lambda, \xi, \alpha, \beta)$ is the Lagrange function.

$$
\begin{cases}
\begin{cases}
\begin{aligned}
f(1) =\ & \max\{\min[u_{ONE}(r_1), u_{ZERO}(r_2), u_{ONE}(r_3)], \\
& \min[u_{ONE}(r_1), u_{ZERO}(r_2), u_{TWO}(r_3)], \\
& \min[u_{TWO}(r_1), u_{ZERO}(r_2), u_{ONE}(r_3)], \\
& \min[u_{TWO}(r_1), u_{ZERO}(r_2), u_{TWO}(r_3)]\} \\
f(2) =\ & \min[u_{ONE}(r_1), u_{ZERO}(r_2), u_{TWO}(r_3)] \\
f(3) =\ & \max\{\min[u_{ZERO}(r_1), u_{ZERO}(r_2), u_{ONE}(r_3)] \\
& \min[u_{ZERO}(r_1), u_{ZERO}(r_2), u_{TWO}(r_3)], \\
& \min[u_{ZERO}(r_1), u_{TWO}(r_2), u_{ONE}(r_3)], \\
& \min[u_{ZERO}(r_1), u_{TWO}(r_2), u_{TWO}(r_3)]\} \\
f(4) =\ & \max\{\min[u_{ZERO}(r_1), u_{TWO}(r_2), u_{ONE}(r_3)], \\
& \min[u_{ZERO}(r_1), u_{TWO}(r_2), u_{ONE}(r_3)]\} \\
f(5) =\ & \min[u_{ZERO}(r_1), u_{ZERO}(r_2), u_{ZERO}(r_3)] \\
& f'(i) = \frac{f(i)}{\sum\limits_{j=1}^{5} f(j)}, i = 0 \sim 5 \\
& y = \mathrm{argmax}(f'(i))
\end{aligned}
\end{cases} & p_i \leq 0.5 \\[2pt]
\begin{cases}
\min \Phi(\omega, \xi) = \frac{1}{2}\|\omega\|^2 + C\sum\limits_{i=1}^{l} \xi_i \\
s.t. \begin{cases} y_i\left[\omega^T \varphi(x_i) + \lambda\right] \geq 1 - \xi_i \\ \xi_i \geq 0, \ i = 1, 2, \ldots, l \end{cases} \\
L(\omega, \lambda, \xi, \alpha, \beta) = \Phi(\omega, \xi) - \\
\sum\limits_{i=1}^{l} \alpha_i\{y_i[\omega^T \varphi(x_i) + \lambda] - 1 + \xi_i\} - \sum\limits_{i=1}^{l} \beta_i \xi_i \\
y = sign[\sum\limits_{i=1}^{l} \alpha_i y_i K(x, x_i) + \lambda]
\end{cases} & p_i > 0.5
\end{cases}
\tag{9}
$$

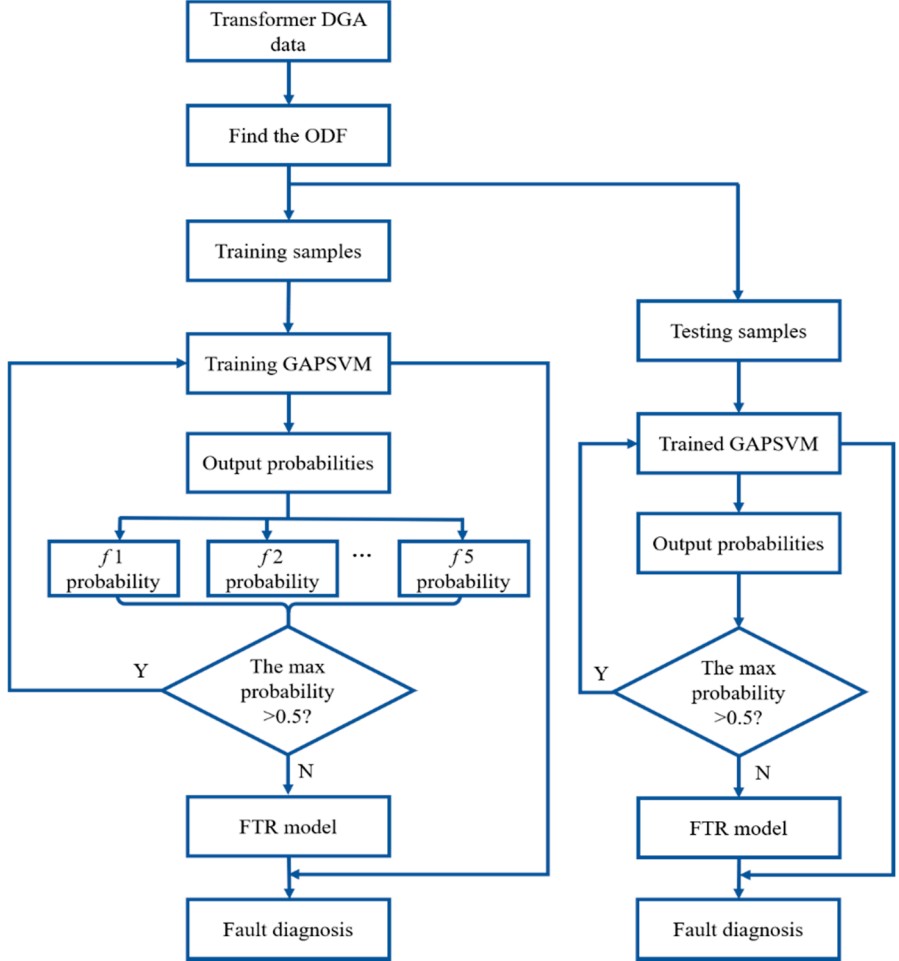

**Figure 2.** Flowchart of GAPSVM integrated with FTR.

## 3. Result Analysis

### 3.1. Fault Sample Data Source and Data Preprocessing

IEC TC 10 is a standard benchmarking dataset for power transformer diagnosis. In total, 118 samples of IEC TC 10 fault data have been randomly divided into training and testing datasets in each computation. The training set includes 93 samples of fault data and the testing set contains 25 samples of fault data. The information of the 118 samples is shown in Table 5.

**Table 5.** Transformer fault sample information.

| Fault Type | LE-D | HE-D | LM-T | H-T | N-C |
|---|---|---|---|---|---|
| Sample quantity | 23 | 45 | 10 | 14 | 26 |

LE-D, HE-D, LM-T, H-T, and N-C, respectively, represent low energy discharge, high energy discharge, low and medium temperature fault, and normal condition. In order to eliminate the error caused by large data variation, the DGA data are normalized by the following equation:

$$x_{ni} = \frac{x_i - x_{i\min}}{x_{i\max} - x_{i\min}} \tag{10}$$

where $x_i$ is the $i$-th sample to be normalized, $x_{i\max}$, $x_{i\min}$ is the maximum and minimum of values before normalization. $x_{ni}$ is the normalized value.

### 3.2. DGA Feature Optimization Result Analysis

After 50-time GA optimal feature selection, DGA features are screened according to CV accuracy. The best three sets of DGA features and their CV accuracy are shown in Table 6. The CV accuracy of No.1 DGA feature (89.83%) is higher than those of No.2 (88.98%) and No.3 (88.14%), so the No.1 DGA feature is considered as the ODF.

**Table 6.** Best three sets of DGA feature.

| DGA Feature | 1 | 2 | 3 |
|---|---|---|---|
| DGA ratios | $H_2/CH_4$ | $H_2/C_2H_4$ | $H_2/C_2H_6$ |
| | $H_2/C_2H_6$ | $H_2/C_2H_6$ | $CH_4/C_2H_2$ |
| | $H_2/TH$ | $H_2/TH$ | $CH_4/C_2H_6$ |
| | $CH_4/C_2H_2$ | $CH_4/CO$ | $C_2H_2/C_2H_4$ |
| | $CH_4/C_2H_6$ | $CH_4/CO_2$ | $C_2H_2/CO$ |
| | $C_2H_2/C_2H_4$ | $C_2H_2/C_2H_4$ | $C_2H_4/TH$ |
| | $C_2H_4/TH$ | $C_2H_2/C_2H_6$ | $C_2H_6/TH$ |
| | $C_2H_6/TH$ | $CO/CO_2$ | $CO/CO_2$ |
| | $CO/CO_2$ | $C_2H_4/TH$ | – |
| | – | $C_2H_6/TH$ | – |
| CV accuracy | 89.83% | 88.98% | 88.14% |

In the process of GA optimal feature selection, the fitness curve of GA is shown in Figure 3, and accuracy for all sample points and the optimal point found by GA is shown in Figure 4. Figure 4a shows the training accuracy of different $c$ and $\sigma$. Figure 4b is the top view of Figure 4a, and the optimal point found by GA is marked in this figure.

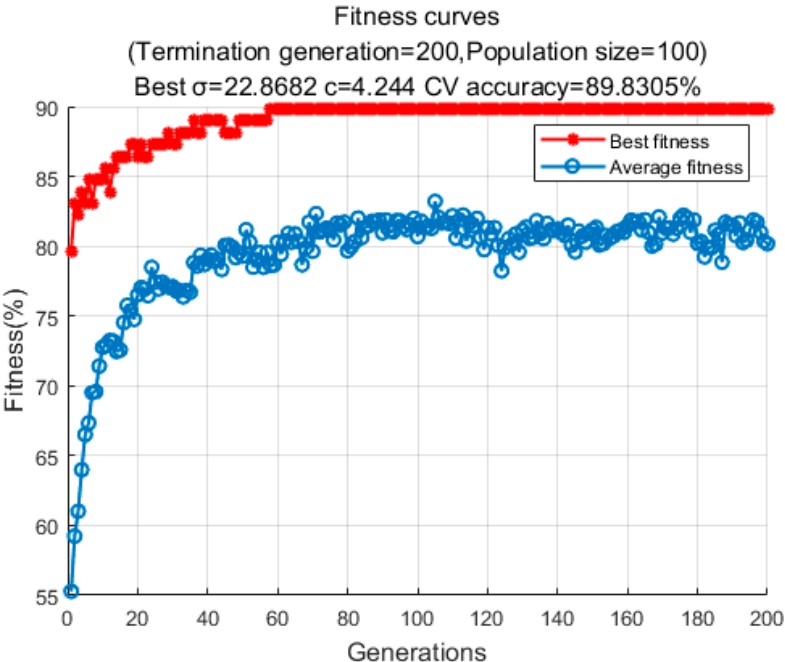

**Figure 3.** Fitness curves of genetic algorithm.

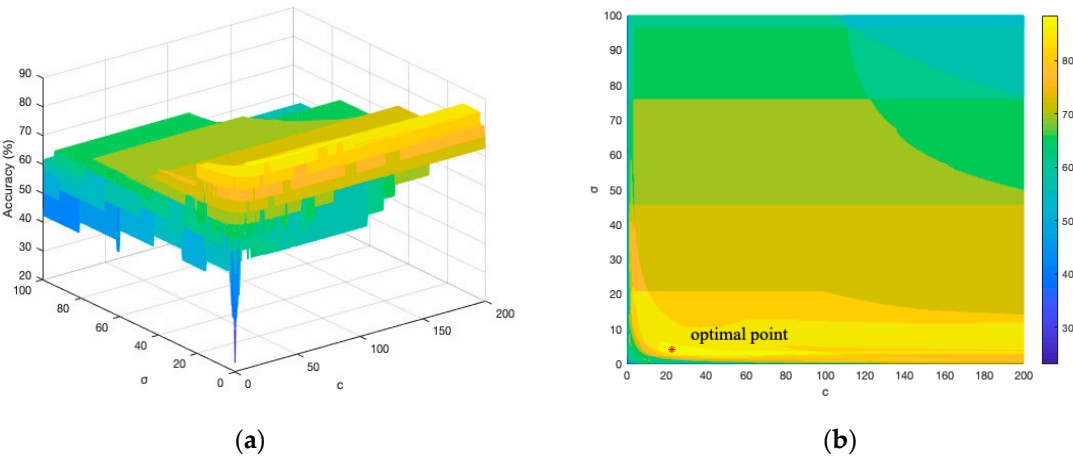

**Figure 4.** Testing accuracy for all *c* and *σ* points. (**a**) 3D visualization of all *c* and *σ* and its corresponding testing accuracy. (**b**) Top view of (**a**).

In order to compare the accuracy between different DGA features, the input features of GAPSVM are divided into three categories: (1) the DGA full data, including $H_2$, $CH_4$, $C_2H_2$, $C_2H_4$, $C_2H_6$, CO, $CO_2$, and TH; (2) the IEC three-ratio feature including $CH_4/H_2$, $C_2H_4/C_2H_6$, and $C_2H_2/C_2H_4$; (3) the ODF including $H_2/CH_4$, $H_2/C_2H_6$, $H_2/TH$, $CH_4/C_2H_2$, $CH_4/C_2H_6$, $C_2H_2/C_2H_4$, $C_2H_4/TH$ $C_2H_6/TH$, $CO/CO_2$. After 30 repeated genetic algorithm optimized SVM (GASVM) calculations, the accuracy of the training and testing sets of the three DGA features is shown in Table 7.

**Table 7.** Accuracy and computing time of each DGA feature.

| Features | Average Accuracy (%) | | Computing Time (s) | |
|---|---|---|---|---|
| | Training | Testing | Training | Testing |
| DGA full data | 89.71 | 57.53 | 37.1847 | $2.19 \times 10^{-4}$ |
| Three-ratio feature | 91.25 | 75.41 | 36.6727 | $1.44 \times 10^{-4}$ |
| ODF | 94.84 | 82.96 | 37.7412 | $2.65 \times 10^{-4}$ |

Both the training and testing accuracy of ODF are higher than those of the other two DGA features, which indicates that the ODF significantly improves the training and testing accuracy of fault diagnosis. Moreover, ODF did not significantly increase the time complexity.

### 3.3. Analysis of the Output of GAPSVM

3.3.1. Threshold Optimization of the Integrated Model

The authors of [24] proposed that when the output probability of PSVM is approximately 0.5, then the sample is near the decision boundary. For the research in this paper, the question of how to find the optimal threshold to determine whether to choose GAPSVM or FTR for diagnosis is of great importance. When the threshold selected is larger, most of the samples will be diagnosed by FTR; when the threshold selected is smaller, most of the samples will be diagnosed by GAPSVM. Hence, choosing the right threshold is essential to the accuracy of the model. Thus, this paper selects nine values from 0.3 to 0.7 in steps of 0.05 as the thresholds to be selected. The training set and the testing set are randomly selected for 10 repeated calculations; the threshold with the highest average fault diagnosis accuracy of the testing set in the 10 repeated calculations is the optimal threshold. The average diagnostic accuracy of each threshold is shown in Figure 5.

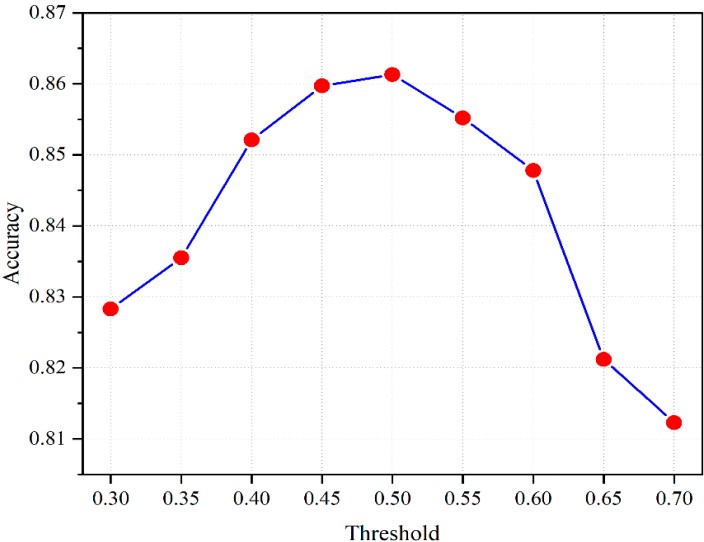

**Figure 5.** Testing accuracy of different thresholds.

It can be seen from the figure that when the threshold is 0.5, the testing accuracy is the highest, because when the threshold is too small, a large number of samples near the decision boundary still choose GAPSVM for diagnosis, but GAPSVM has a lower diagnostic accuracy for samples at the decision boundary; when the threshold selected is too large, a large number of samples that are not near the decision boundary are diagnosed by FTR. For samples that are not near the decision boundary, the diagnostic accuracy of FTR is lower than that of GAPSVM. Moreover, the balance is reached when the threshold is selected as 0.5, so the optimal threshold is chosen as 0.5.

### 3.3.2. Analysis of Accuracy of GAPSVM

The training and testing accuracy of the maximum output probability which are larger and equal to 0.5 or smaller after 30 repeated GAPSVM calculations, with the ODF feature as input, are listed in Table 8.

**Table 8.** Accuracy of max probability >0.5 and max probability ≤0.5 samples.

| Max Probability | | Training Accuracy | Testing Accuracy |
|---|---|---|---|
| | Max | 100% | 88.64% |
| >0.5 | Min | 97.08% | 82.96% |
| | Mean | 97.53% | 86.85% |
| | Max | 92.00% | 85.71% |
| ≤0.5 | Min | 85.00% | 50.00% |
| | Mean | 89.83% | 56.72% |

It can be identified from Table 8 that the training and testing accuracy of the maximum probability >0.5 is much higher than that of the maximum probability ≤0.5, which reflects the deficiency of the GAPSVM when a sample's maximum probability ≤0.5. As described in Section 2, the FTR model will be applied to diagnose the samples of max probability ≤0.5; the training set and testing set accuracy of the FTR model is 76.76% and 75.25%, respectively. The FTR model significantly improves the accuracy of the testing set and does not reduce the accuracy of the training set, although its accuracy is less than that of the SVM in the training samples with the max probability ≤0.5, because the average number of samples in the training set with a max probability ≤0.5 is only 1.7 in 30 calculations.



### 3.4. Comparisons with Other Diagnosis Methods

Back propagation neural network (BPNN), K-Nearest Neighbor (kNN), and GASVM are usually used in traditional power transformer fault diagnosis, when ODF is adopted as the input feature of these methods. The testing accuracy of the above methods and two published studies is listed in Table 8 and the accuracy of 10-time computation of different methods is shown in Figure 6.

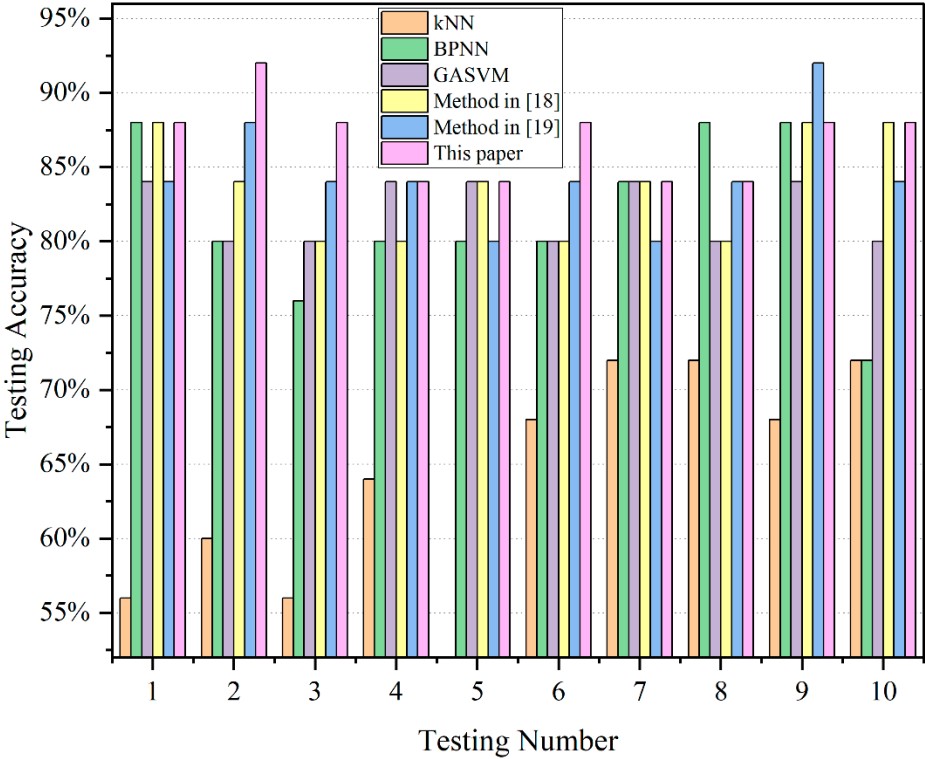

**Figure 6.** Testing accuracy of different methods.

As shown in Table 9, the testing accuracy of the GAPSVM integrated with the FTR model proposed in this paper reaches 86.80%, which is higher than that of kNN (64.00%), BPNN (81.60%), and GASVM (82.00%), the method in [18] (83.60%) and the method in [19] (84.40%). It can be also seen from Figure 6 that, in most cases, this model performs better than the traditional methods.

**Table 9.** Average testing accuracy of different methods.

| Diagnosis Method | Testing Accuracy (%) |
|:---:|:---:|
| kNN | 64.00 |
| BPNN | 81.60 |
| GASVM | 82.00 |
| Method in [18] | 83.60 |
| Method in [19] | 84.40 |
| This Paper | 86.80 |

### 3.5. Model Evaluation

To verify the validity and generalization ability of the proposed model, 30 groups of DGA fault samples from the State Grid Co. of China are used as testing samples of the trained model in 3.4; the diagnostic results are shown in Table 10.

**Table 10.** Diagnostic results of 30 groups of DGA fault data.

| Fault Type | LE-D | HE-D | LM-T | H-T | N-C |
|---|---|---|---|---|---|
| True samples | 6 | 6 | 5 | 5 | 7 |
| Predicted samples | 6 | 6 | 7 | 6 | 4 |

From the diagnostic results of the 30 DGA samples, the proposed model is able to correctly classify 26 samples, and the accuracy can reach 86.67%. Furthermore, confusion matrix, F-measure, precision, and recall are introduced to examine the performance of the proposed model. The confusion matrix illustrates the relationship between predicted fault types and true fault types. Precision indicates the percentage of the samples that are identified as positive categories which are indeed positive categories, while the recall indicates the percentage of the positive examples which are predicted correctly in the dataset. On the other hand, F-measure is a weighted harmonic average of precision and recall, which provides a single score that balances both the concerns of precision and recall in one number. Equations of each measure index are shown as follows.

$$precison = \frac{TP}{TP + FP} \tag{11}$$

$$recall = \frac{TP}{TP + FN} \tag{12}$$

$$F_{measure} = 2 \times (\frac{precision \times recall}{precision + recall}) \tag{13}$$

It can by identified from Table 11 that the model can effectively diagnose most of the fault types. Precision, recall, and F-measure are 0.875, 0.874, and 0.859, respectively. The above measure indexes and confusion matrix proved the validity and generalization of proposed model.

**Table 11.** Confusion matrix of the diagnostic result.

| | | Predicted by the Proposed Model | | | | |
|---|---|---|---|---|---|---|
| | | LE-D | HE-D | LM-T | H-T | N-C |
| | LE-D | 6 | 0 | 0 | 0 | 0 |
| | HE-D | 0 | 7 | 0 | 0 | 0 |
| Actual | LM-T | 0 | 0 | 4 | 1 | 0 |
| | H-T | 0 | 0 | 0 | 5 | 0 |
| | N-C | 0 | 1 | 2 | 0 | 4 |

*3.6. Model Validation Using Practical Dataset*

In order to verify the performance of the method proposed in this article in practical applications and other datasets, the dataset of [18] is cited. The lack of some DGA data in the actual operation of the transformers is considered in this dataset, in which one or two gases are null. The information of the dataset is shown in Table 12. Firstly, the missing dissolved gas is replaced by the average value of the gas corresponding to the fault type. Because $C_2H_6$ in HE-D are all missing values, the $C_2H_6$ value of HE-D is replaced by the average value of $C_2H_6$ gas corresponding to the fault type in the IEC TC 10 database. Then, kNN, BPNN, GASVM, the method in [18], the method in [19], and the model proposed in this paper are used to diagnose the fault types of DGA samples. The fault types of DGA samples diagnosed by this method are shown in Table 12. Moreover, the diagnostic accuracy of the different methods is shown in Table 13.

**Table 12.** DGA data information of [18].

| Actual Fault | $H_2$ | $CH_4$ | $C_2H_2$ | $C_2H_4$ | $C_2H_6$ | CO | $CO_2$ | TH | Diagnostic Result |
|---|---|---|---|---|---|---|---|---|---|
|  | 78 | 20 | 28 | 13 | 11 | / | 784 | 72 | LE-D |
| LE-D | 95 | 10 | 39 | 11 | / | 122 | 467 | 60 | LE-D |
|  | 8 | / | 101 | 43 | / | 192 | 4067 | 144 | LE-D |
|  | 7020 | 1850 | 4410 | 2960 | / | 2140 | 1000 | 9220 | HE-D |
| HE-D | 120 | 31 | 94 | 66 | / | 48 | 271 | 191 | HE-D |
|  | 5100 | 1430 | 1010 | 1140 | / | 117 | 197 | 3580 | HE-D |
|  | 48 | 610 | / | 10 | 29 | 1900 | 970 | 649 | HE-D |
| LM-T | 12 | 18 | / | 4 | 4 | 559 | 1710 | 26 | LM-T |
|  | 66 | 60 | / | 7 | 2 | 76 | 90 | 69 | LM-T |
|  | 8800 | 64,064 | / | 95,650 | 72,128 | 290 | 90,300 | 231,842 | H-T |
| H-T | 1100 | 1600 | 26 | 2010 | 221 | / | 1430 | 3857 | H-T |
|  | 1860 | 4980 | 1600 | 10,700 | / | 158 | 1300 | 17,280 | LM-T |
|  | 134 | 134 | / | 45 | 157 | 1008 | 10,528 | 336 | H-T |
| N-C | / | 225 | 3 | 110 | 225 | 785 | 4500 | 563 | N-C |
|  | 200 | 3 | / | 200 | 50 | 1000 | 20,000 | 253 | N-C |

**Table 13.** Fault types diagnosed by proposed model and other algorithms.

| Algorithms | kNN | BPNN | GASVM | Method in [18] | Method in [19] | This Paper |
|---|---|---|---|---|---|---|
| Accuracy | 66.67% | 73.33% | 73.33% | 80.00% | 80.00% | 86.67% |

It can be identified from the diagnostic results that the integrated model proposed in this paper is able to correctly diagnose 13 of 15 DGA fault samples. The fault diagnosis accuracy reached 86.67%, which is higher than kNN (66.67%), BPNN (73.33%), GASVM (73.33%), the method in [18] (80%), and the method in [19] (80%). The diagnostic results proved the superiority and robustness of the integrated model.

## 4. Conclusions

In this paper, GA combined with SVM is used to select the ODF, which is adopted as the input feature of the proposed fault diagnosis model. Aiming at eliminating the insufficiency of GASVM in some samples which are located near the decision boundary, an AI and expert experience combined model based on the GAPSVM integrated with FTR is proposed, which is the main innovation of this paper. The conclusions are as follows:

- The ODF is selected from 36 DGA features by the GA and SVM, and the average testing accuracy of GASVM is 82.96%, which is higher than that of the IEC three-ratio feature (75.41%) and DGA full data (57.53%). The ODF is more suitable as the input feature of the power transformer fault diagnosis model.
- The AI and expert experience combined model is established based on the IEC TC 10 dataset, and the average testing accuracy is 86.80% after 10-time computation, which is higher than kNN (64.00%), BPNN (81.60%), GASVM (82.00%), the method in [18] (83.60%), and the method in [19] (84.4%). Specifically, this model avoids misclassification efficiently when a sample is near the decision boundary of GAPSVM. Moreover, when 30 groups of DGA data from the State Grid Co. of China are diagnosed by the proposed model trained by 118 groups of IEC TC 10 DGA data, diagnostic accuracy is 86.67%. Additionally, the validity and generalization are verified by measure indexes of classification.
- A total of 15 real cases with missing values are tested by six methods. GAPSVM integrated with the FTR model correctly diagnosed the fault types of the 13 cases, which proves that AI-based algorithms integrated with expert experience have great robustness.

**Author Contributions:** Conceptualization, Y.Z. and Y.W.; methodology, Y.Z.; software, Y.W.; validation, X.F.; formal analysis, X.F.; investigation, Z.S.; resources, Z.S.; data curation, Y.Z.; writing—original draft preparation, Y.W.; writing—review and editing, Y.Z. and Z.S.; visualization, W.Z.; supervision, R.Z. and J.H.; project administration, Y.Z.; funding acquisition, Y.Z. All authors have read and agreed to the published version of the manuscript.

**Funding:** This research was funded part by the National Natural Science Foundation of China under Grant 61473272 and Grant 51867003, the Natural Science Foundation of Guangxi (2018JJB160056; 2018JJB160064; 2018JJA160176), the Guangxi Thousand Backbone Teachers Training Program, The Boshike Award Scheme For Young Innovative Talents, the Basic Ability Promotion Project for Yong Teachers in Universities of Guangxi (2019KY0046; 2019KY0022) and the Guangxi Bagui Young Scholars Special Funding in support of this work.

**Conflicts of Interest:** The authors declare no conflict of interest.

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
