# Peer review of "An Integrated Model for Transformer Fault Diagnosis to Improve Sample Classification near Decision Boundary of Support Vector Machine"

_energies, doi:10.3390/en13246678_

Round 1

Reviewer 1 Report

This manuscript presents a GA based SVM classifier to optimize the parameters of the SVM. However, using evolutionary optimization algorithms for model development in DGA fault diagnosis is not novel. The author has also combined the probabilistic SVM model with a fuzzy approach to enhance the performance of their model. The topic of the manuscript is interesting and is of high interest for industry and researchers. However, the novelty of the manuscript is not clear and there is a complete lack of comparison with the published state-of-the-art models. In addition, it is difficult for the readers to follow the story line of the manuscript. The authors should address the below comments and revise the manuscript extensively before considering it for publication in Energies. Please see the below comments for more details.

  1. As mentioned above it is difficult to follow and understand some parts of the manuscript especially Abstract and Introduction sections. The authors are highly recommended to rewrite these sections and modify other sections too.
  2. Although the authors have reviewed some related papers, there are some state-of-the-arts that are missing such as: 
    • Ma, Hui, et al. "Smart transformer for smart grid—intelligent framework and techniques for power transformer asset management." IEEE Transactions on Smart Grid 6.2 (2015): 1026-1034.
    • Peimankar, Abdolrahman, et al. "Evolutionary multi-objective fault diagnosis of power transformers." Swarm and Evolutionary Computation 36 (2017): 62-75.
    • Yang, Z., et al. "Association rule mining-based dissolved gas analysis for fault diagnosis of power transformers." IEEE Transactions on Systems, Man, and Cybernetics, Part C (Applications and Reviews) 39.6 (2009): 597-610.
    • Ashkezari, Atefeh Dehghani, et al. "Application of fuzzy support vector machine for determining the health index of the insulation system of in-service power transformers." IEEE Transactions on Dielectrics and Electrical Insulation 20.3 (2013): 965-973.
    • Peimankar, Abdolrahman, et al. "Ensemble classifier selection using multi-objective PSO for fault diagnosis of power transformers." 2016 IEEE Congress on Evolutionary Computation (CEC). IEEE, 2016.
    • Shintemirov, Almas, Wenhu Tang, and Q. H. Wu. "Power transformer fault classification based on dissolved gas analysis by implementing bootstrap and genetic programming." IEEE Transactions on Systems, Man, and Cybernetics, Part C (Applications and Reviews) 39.1 (2008): 69-79.
  3. In Section 2.1.2, the authors should explain the GA encoding better especially for C and σ. For example, what is the size and range?
  4. In Section 2.2.1, there is not a good justification on why triangular membership function has been used.
  5. In Section 2.2.2, a threshold equal to 0.5 is used to classify the samples. However, there is not experimental validation on why this threshold is the best. The authors should use other values and report the effectiveness on the model classification capability.
  6. Figure 2 should be redrawn. Because it is not clear what happens to the test data. In addition, after the condition (if p>0.5), it is not very clear why there is loop back to the GASVM? Should not it be the output of the model?
  7. There is not enough explanation about Equation 5. The authors have to explain all the parameters clearly.
  8. Figure 5 should be modify. The line styles should be changed to be distinguishable. It is even better to replace the figure with a bar plot.
  9. The author should report the time complexity of their model and compared with other approaches in Table 6.
  10. As mentioned above, there is complete lack of comparison with other published papers using DGA for power transformers fault diagnosis. The authors have to add a comparison section and compare their model with other state-of-the-arts such as the ones listed above and cited in the manuscript.
  11. The author should a table about the distribution of samples per each fault class for the IEC TC 10 dataset. If the class distribution are not balanced, the author should make the dataset balanced by using oversampling methods such as Synthetic Minority Oversampling Technique (SMOTE) and compare the performance of their model before and after oversampling.
  12. The authors should validate their model using other dataset used in the literature such as the one in (Li, Enwen, Linong Wang, and Bin Song. "Fault diagnosis of power transformers with membership degree." IEEE Access 7 (2019): 28791-28798) or other papers.  
  13. The explanations/definitions of "precision" and "recall" are not accurate in the text.
  14. Please modify/replace Figure 6-b to a more descriptive figure/table.
  15. Please replace the word "groups" in Section 3.1 with "samples". 

Reviewer 2 Report

The paper is interesting and well written.

However, the methods applied in solving the presented problem might be too specific for the common reader of Energies. Hence, I warmly suggest to add a short paragraph reporting some simple and general explanation of SVM and any other related topics that can help the reader in better understanding the paper content.

Reviewer 3 Report

In this paper, the authors presented an SVM-based method to diagnose transformer fault based on DGA.

In my opinion:

a)Eq 2 should be corrected due to a broken font

b)In part 3.6 it is said that inter-turn short circuit leads to H-T fault for transformer 2. However, data in Tables 11 and 12 suggest HE-D fault for your method. No method predicted H-T fault.

c) I do not think that using two transformers for evaluation (one even in N-C) is good enough to conclude that the algorithm works/not works. In my opinion, the sample should be considerably larger. 

Round 2

Reviewer 1 Report

I would like to thank the authors for addressing my comments. The quality of the manuscript has been improved substantially. 

Reviewer 2 Report

The authors have sufficiently addressed the previous remarks.

Reviewer 3 Report

no further comments.